# Multi-Perspective Observation on the Prevalence of Food Allergy in the General Chinese Population: A Meta-Analysis

**DOI:** 10.3390/nu14235181

**Published:** 2022-12-06

**Authors:** Jian Wang, Wenfeng Liu, Chunyan Zhou, Fangfang Min, Yong Wu, Xin Li, Ping Tong, Hongbing Chen

**Affiliations:** 1State Key Laboratory of Food Science and Technology, Nanchang University, Nanchang 330047, China; 2School of Food Science and Technology, Nanchang University, Nanchang 330031, China; 3Sino-German Joint Research Institute, Nanchang University, Nanchang 330047, China; 4Jiangxi Province Key Laboratory of Food Allergy, Nanchang University, Nanchang 330047, China; 5Department of Translational Medicine Institute, Shanghai Jiao Tong University, Shanghai 200127, China

**Keywords:** food allergy, prevalence, China, meta-analysis

## Abstract

Reliable estimates of the prevalence of food allergy (FA) among the general Chinese population have remained unclear. This meta-analysis aims to provide an accurate estimate of FA prevalence in China with comprehensive data. A systematic literature search was conducted in eight electronic databases, i.e., China National Knowledge Infrastructure, Wanfang, Weipu, China Biology Medicine, Web of Science, PubMed, Cochrane, and Embase. A random-effects model was used to analyze the pooled prevalence of FA for four different assessment methods. A total of 46 eligible articles were included in the narrative synthesis, and 41 articles were ultimately included in the meta-analysis. Overall, the pooled prevalence of self-reported FA, self-reported physician-diagnosed FA, SPT (specific skin prick test)-positive FA, and OFC (open food challenge)-positive FA were 11.5% (95% CI: 9.8–13.5%), 5.3% (95% CI: 4.2–6.5%), 11.6% (95% CI: 9.6–14.1%), and 6.2% (95% CI: 4.4–8.7%), respectively. Subgroup analyses suggested that the prevalence of FA was affected by age, year of data collection, region, and sample size, but not by gender. This meta-analysis indicated that FA is common among Chinese people, with an increasing trend in prevalence during the past two decades. Given the high heterogenicity between these studies, a national survey with a large sample size based on standardized diagnosis is urgently needed to gain a more scientific understanding of the actual situation of food allergy in China.

## 1. Introduction

Food allergy (FA) is an abnormal immune response to certain foods, which can trigger a series of clinical symptoms (from mild forms with organ localization to severe and potentially fatal conditions with systemic involvement) after reproducible exposure to a specific food [1]. It is well-known that the incidence and development of FA result from complex interactions of genome-environment-lifestyle factors, and FA as the “second wave” of the allergy epidemic has become an increasing global health concern and imposed a serious burden on modern society [2,3]. However, an effective cure and prevention for FA currently are still not available. Thus, it is crucial to get the epidemiological characteristics of FA to help better understand its pathogenesis and development, which can further provide strategies for possible preventive interventions and treatment for FA.

The double-blind placebo-controlled food challenge (DBPCFC) is the golden standard for FA diagnosis, but it is time-consuming, resource-intensive, and risk-filled. Therefore, self-reported clinical history, food-specific skin prick test (SPT), food-specific serum IgE (sIgE), and open food challenge (OFC) were widely used as FA diagnostic tools in many epidemiologic studies. Due to the difference in FA diagnostic tools employed in the epidemiological investigations of FA, the robust data on FA prevalence remain elusive, including in China. Located in eastern Asia, China is one of the largest countries in the world, with a vast territory of 9.6 million square kilometers and a multiethnic population of more than 1.4 billion. Accordingly, understanding the current situation of FA in China is deemed essential to global public health. At present, certain studies have reported the prevalence of FA in China, with the prevalence varying widely from 3.13% to 21.13% [4]. Likewise, given the inconsistency in survey periods, population characteristics, studies region, and diagnostic criteria among these studies, there is an urgent need for evidence-based strategies to estimate the prevalence of FA in the general Chinese population.

The National Academy of Sciences (NAS) report provides a comprehensive review of global FA epidemiologic studies and demonstrates that the limitations of these prevalence data make it challenging to derive definitive global statistics [5]. Nevertheless, systematic evaluations or meta-analysis are still able to provide valuable information on potential risk factors for FA prevalence and insights into variability based on study populations and methods. Moreover, the prevalence of FA would be categorized into point prevalence (the proportion of the population suffering from food allergy at a specific time), period prevalence (the proportion of the population suffering from food allergy during a given period), and lifetime prevalence (the proportion of the population experiencing food allergy at some point in their lives) according to different epidemiological measures [6]. Based on the above, we have conducted this meta-analysis to estimate the prevalence of FA in the general Chinese population, which will provide a more accurate and comprehensive understanding of the prevalence of FA in China.

## 2. Methods

This meta-analysis was registered at PROSPERO (CRD42022315941) and conducted according to the Preferred Reporting Items for Systematic Reviews and Meta-Analyses (PRISMA) [7].

### 2.1. Search Strategy

Potential studies were thoroughly searched using eight electronic databases, including four Chinese-language databases (China National Knowledge Infrastructure (CNKI), Wanfang, Weipu, China Biology Medicine (CBM)) and four English-language databases (Web of Science, PubMed, Cochrane, Embase) from inception date until August 2022. The terms considered in the search strategy were as follows: “food hypersensitivity” or “food allergy”; “prevalence” or “frequency” or “epidemiology” or “incidence” or “survey” or “rate”; “China” or “Chinese” or “Hong Kong” or “Macao” or “Taiwan” or “Mainland”. The language of publications was restricted to English and Chinese.

### 2.2. Inclusion and Exclusion Criteria

Inclusion criteria for the study design: (1) the prevalence of FA was investigated among the general Chinese population; (2) cross-sectional studies, cohort studies, or comparative studies; (3) studies with data on the total number of participants and number of cases of food allergy or prevalence of food allergy; (4) diagnostic method based on the questionnaire, SPT sensitization, sIgE sensitization, and food challenge. Exclusion criteria for study design: (1) several articles were studying the same participants, the one with more detailed data would be selected; (2) review papers or conference and meeting abstracts; (3) studies conducted in special populations; (4) studies that do not clearly report the prevalence of food allergy and for which the relevant content cannot be extracted; (5) low evaluation score (risk of bias assessment).

### 2.3. Data Extraction

After independently screening titles and abstracts, two reviewers independently retrieved and reviewed the full-text studies based on the inclusion and exclusion criteria. The following information was independently extracted from each study: name of the first author, publication year, survey period, study region, sample size, participant characteristics, number of food allergy cases, and the prevalence of food allergy. Any discrepancies in these processes were discussed and identified with a third reviewer to reach a consensus.

### 2.4. Quality Assessment

The quality of included studies was independently evaluated by two reviewers using the relevant version of the Joanna Briggs Institute (JBI) quality assessment tool. The checklists consisted of 9 items. Each item was graded as yes, no, or unclear. A “yes” response was scored as 1, while “no” and “unclear” responses were scored as 0. The total score ranged from 0–9. Studies rated 7–9 were considered “high quality”, those rated 5–7 were “moderate quality”, and those rated 0–4 were “low quality”. High- and medium-quality studies were included for meta-analysis. Any discrepancies were resolved through discussion or arbitration by a third reviewer.

### 2.5. Data Analysis

Statistical analyses were performed using the R program version of 4.2.0. The pooled prevalence of FA and 95% confidence intervals (CI) were calculated. The Q and I² tests were employed to evaluate the heterogeneity of the studies. A random-effects meta-analysis model was used when the heterogeneity was statistically significant (I² ≥ 50%, *p*≤ 0.05); otherwise, the fixed-effects model was used. Subgroup analyses, sensitivity analyses, and meta-regression analyses were performed to analyze the sources of heterogeneity and factors potentially influencing the prevalence of FA. Finally, Egger’s test and funnel plots were applied to explore potential publication bias.

## 3. Results

### 3.1. Inclusion of Studies and Characteristics

Figure 1 is a PRISMA flowchart for screening articles. Initially, 7023 articles were identified through electronic databases. After the removal of duplicates (n = 2083) and reviews (n = 305), 4635 articles’ titles and abstracts were checked, but only 145 articles were left for the full-text assessment. Eventually, 46 studies that met the criteria were included for the final assessment.

Table 1 lists the general characteristics of the 46 articles. We classified the prevalence based on the FA diagnosis tools used in these studies, i.e., questionnaires (self-report and physician-diagnosed self-report were the most used), SPT sensitization, specific IgE sensitization, and food challenges were employed to assess the prevalence of FA in China. Among them, three articles reporting the point prevalence of sIgE positive FA (28.81%, 3.19%, and 9.06%, respectively) [8,9,10] were not included for a pooled prevalence estimate due to significant heterogeneity; the two articles that only described the period/point prevalence of self-reported FA without providing lifetime prevalence data were also not applicable to the meta-analysis [11,12]. Finally, 41 studies were included in the meta-analysis to estimate the pooled lifetime prevalence of self-reported FA, self-reported physician-diagnosed FA, SPT-positive FA, and OFC-positive FA.

The quality of the 41 studies included for the meta-analysis was evaluated by the JBI checklist (9 items). The results (Appendix A) show that all studies were rated as “moderate quality” or “high quality”.

### 3.2. Pooled Prevalence of FA

The estimated pooled lifetime prevalence of self-reported FA reported in 29 studies was 11.5% (*n* = 465,330, 95% CI: 9.8–13.5%), and self-reported physician-diagnosed FA in 17 studies was 5.3% (*n* = 109,741, 95% CI: 5.3–6.5%). Based on the nine studies with available data, the pooled point prevalence of SPT-positive FA was 11.6% (*n* = 8815, 95% CI: 9.6–14.1%). The 11 included studies revealed that the pooled point prevalence of OFC-positive FA was 6.2% (*n* = 13,059, 95% CI: 4.4–8.7%). All values of the pooled prevalence are shown in Figure 2.

### 3.3. Subgroup Analyses

The pooled prevalence of FA among all subgroups according to gender, age, year of data collection, regions, and sample size are summarized in Table 2. Further details are available in Appendix A.

Prevalence of FA by gender. There was no significant difference in the lifetime prevalence of FA between males and females, with a rate of 10.7% (95% CI: 8.2–13.8%) for males and 10.2% (95% CI: 7.8–13.3%) for females by the self-report assessment method, with a rate of 4.6% (95% CI: 3.5–5.9%) for males and of 4.0% (95% CI: 2.9–5.4%) for females by the self-reported physician-diagnosis assessment method. The prevalence data of SPT- and OFC-positive FA by gender were not available.

Prevalence of FA by age. Subgroup analyses showed that the lifetime prevalence of FA increased with age. The lifetime prevalence of self-reported FA rose from 10.5% (95% CI: 7.7–14.2%) in those aged 0–3 years to 14.2% (95% CI: 12.2–16.5%) in those above 18 years of age, while the lifetime prevalence of self-reported physician-diagnosed FA was highest (7.2%, 95% CI: 6.2–8.2%) among those above 18 years old and lowest (4.0%, 95% CI: 2.1–7.4%) among those less than 3 years old. The age subgroup analysis could not be performed on the point prevalence of SPT- and OFC-positive FA, both of which focused on populations aged 0 to 3 years.

Prevalence of FA by year of data collection. Subgroup analyses of FA prevalence at each 10-year interval showed that the prevalence appears to be higher in 2011–2021 than in 1999–2010. The lifetime prevalence of self-reported FA was significantly higher in 2011–2021 (12.5%; 95% CI: 10.4–15.0%) than in 1999–2010 (8.7%; 95% CI: 6.8–11.0%). Likewise, the prevalence of self-reported physician-diagnosed FA and OFC-positive FA increased from decade to decade, albeit not significantly.

Prevalence of FA by regions. Subgroup analyses were performed according to four economic regions of China (northeast, east, central, and west). The results showed that the prevalence of FA varied by region, with a higher prevalence in West China ((12.1% (95% CI: 9.5–15.4%) for self-reported FA, 10.0% (95% CI: 6.1–16.0%) for self-reported physician-diagnosed FA, 13.0% (95% CI: 11.2–15.1%) for SPT-positive FA, and 6.9% (95% CI: 5.3–9.0%) for OFC-positive FA). Moreover, studies on the prevalence of FA mainly focused on East and West China. The details about the geographic distribution of included studies on the prevalence of FA in China are shown in Figure 3.

Prevalence of FA by sample size. Regarding the prevalence of FA according to sample size, the prevalence of FA was relatively higher in studies with a smaller sample size (sample size <1000; 16.3% (95% CI: 13.0–20.2%) for self-reported FA and 6.6% (95% CI: 2.1–19.0%) for self-reported physician-diagnosed FA), lower in large sample studies (sample size >5000; 10.0% (95% CI: 7.6–13.1%) for self-reported FA, and 3.6% (95% CI: 2.6–5.1%) for self-reported physician-diagnosed FA). The studies on the prevalence of SPT- and OFC-positive FA were mainly focused on “sample size <1000” (prevalence 13.5% (95% CI: 11.6–15.6%), 6.9% (95% CI: 4.9–9.5%), respectively), and there were no studies with large sample sizes (sample size >5000).

### 3.4. Meta-Regression Analyses

The high heterogeneity was observed in all evaluated studies. Meta-regressions were performed on gender, age, year of data collection, regions, and sample size to explore the potential sources of heterogeneity (Appendix A). The results indicated that sample size could significantly affect the heterogeneity of the prevalence of FA measured by all methods. However, this result did not fully clarify the high level of heterogeneity.

### 3.5. Sensitivity Analysis and Publication Bias

Sensitivity analysis (Appendix A) did not find any individual study that significantly influenced the overall results in the meta-analysis for the prevalence of self-reported and self-reported physician-diagnosed FA, indicating that our statistical results were relatively stable. However, the stability of pooled prevalence of SPT- and OFC-positive FA was influenced by certain articles [37,43], with prevalence rates of 12.6% (95% CI, 11.1–14.4%) and 7.4% (95% CI, 6.1–8.9%), respectively, after excluding the influential articles (Appendix A).

Evidence of substantial publication bias was identified using Egger’s test and funnel plots (Appendix A). In the meta-analysis of self-reported FA prevalence, Egger’s test (t = 3.55, *p*-value = 0.0016) showed significant publication bias, and the funnel plot for these studies had a significant asymmetric visual inspection. To assess a potentially low publication bias, a trim-and-fill method was performed, and then 13 studies were added to the model with a final combined effect size of 6.7% (95% CI, 5.1–8.8%). The results (Egger’s test: *p* > 0.05, funnel plots) suggested that there was no publication bias in the meta-analysis for the prevalence of self-reported physician-diagnosed SPT- and OFC-positive FA.

## 4. Discussion

To the best of our knowledge, this study is the first meta-analysis that incorporates the most comprehensive data to estimate the prevalence of FA among the general Chinese population. Consistent with previous epidemiological studies [54,55], we also found that the prevalence estimates for self-reported FA (11.5%) and SPT-positive FA (11.6%) were higher than the prevalence of self-reported physician-diagnosed FA (5.3%) and OFC-positive FA (6.2%). Self-reports are known to overestimate the true prevalence of FA because these may also include food intolerances or toxicities. Moreover, a positive SPT only represents sensitization and not allergy, which may also exaggerate the prevalence of FA to some extent. Studies in Africa have also found high rates of food sensitization and self-reported food allergy [56]. In this meta-analysis, relatively few studies with food challenge tests for assessments were available for inclusion, as the prevalence estimates for FA in China are mainly based on self-reported reactions to foods, implying an overestimated prevalence of FA.

In a meta-analysis of FA epidemiology in Europe, Nwaru et al. [57] observed an overall lifetime self-reported prevalence of 17.3% for FA and 0.9% for the positive food challenge, which was different from our study. This may be due to the different population distribution of the two studies, i.e., the Chinese research focused mainly on children and rarely on adults. Moreover, the prevalence of FA may vary by region and country due to various economic status, geographical environments, genetic backgrounds, and dietary habits. The 2010 FDA Food Safety Survey showed that 13% of self-reported adults suffered from FA and 6.5% had FA by physician-diagnosis [58], which was consistent with the corresponding prevalence figures in our meta-analysis. Indeed, SPT and sIgE tests merely indicate food sensitization, and an accurate diagnosis for FA generally requires a combination of clinical history. Based on medical history and positive sIgE and SPT results, the EuroPrevall-INCO surveys showed that the prevalence of FA was 1.50% in Hong Kong, China, 0.21% in Guangzhou, China, 0.69% in Shaoguan, China, and 0.14% in India [39]. Although DBPCFC is the gold standard for food allergy diagnosis, OFC is traditionally more commonly used than DBPCFC because of the challenges and limitations of DBCFC in practical applications. The prevalence of OFC-positive FA in children was reported to be 4% in the UK [59], 3.6% in Denmark [60], 6.8% in Norway [61], and 4% in Australia [62].

In our meta-analysis, the prevalence of FA did not differ remarkedly between genders, but significantly differed between age groups. Notably, there were certain studies indicating age-related gender differences in FA [63,64,65]. However, our study did not find sufficient evidence to support or refute this view, so the further in-depth investigation is necessary. Here, we found that the lifetime prevalence of FA was significantly higher in adults than in children, mainly because the cumulative prevalence increased relatively with age and duration of exposure. In general, based on the natural history of FA, the point prevalence of FA decreases with age; in other words, many children outgrow reactions to foods by age [66]. Unfortunately, studies on the point prevalence of SPT- and OFC-positive FA in China are mainly based on young children (0–3 years old).

In our study, the prevalence of FA appears to have increased over the last few decades, although statistically significant differences were found only in self-reported FA prevalence. Comparing data from cross-sectional epidemiological surveys conducted at different time points is a good method to obtain trends of FA prevalence over time. However, conclusions may be influenced by the initial purpose of the study, protocol design, sample characteristics, and diagnostic criteria. Ideally, the epidemiologic trends in FA should be evaluated in the same region over consecutive periods using the same methodology. Three cross-sectional studies were performed ten years apart (1999, 2009, and 2019) using the same survey methods to determine the prevalence of sensitization and challenge-proven FA in children aged 0–2 years at the same hospital clinic in Chongqing, China, and it was found that the prevalence of FA was significantly higher in 2009 (7.7%) and 2019 (11.1%) than in 1999 (3.5%) [48]. Epidemiological studies in developed countries have shown a significant increase in the prevalence of FA, which may be due to environmental exposures, economic development, and lifestyle changes [67]. In the United States, studies have shown that the prevalence of self-reported FA has increased by 1.2% per decade [68]. The hospitalization rate for food-induced allergic reactions in the United Kingdom increased from 1.2 to 2.4 per 100,000 people between 1998 and 2012 [69]. In Australia, the number of hospitalization for food-induced allergic reactions increased by an average of 13.2% per year between 1994 and 2005 [70].

This meta-analysis provides a comprehensive view of the current state of FA in China from different assessment methods (self-report, self-reported physician-diagnosis, SPT sensitization, and OFC sensitization). However, several potential limitations should be carefully considered when interpreting the findings. First, similar to other meta-analysis of epidemiological studies [71], this study had significant heterogeneity, although random-effect models were carried out to obtain conservative prevalence estimates. Second, with the exception of eastern and western China, relatively few studies were conducted in northeastern and central China, and thus the prevalence of FA in these regions may have been underestimated or overestimated. Considering that different socio-cultural and economic factors among these four regions of China (northeast, east, central, and west) may lead to differences in prevalence, this incomplete survey may limit the generalizability of the findings at the national level. Third, most of the included studies, especially those on the prevalence of SPT- and OFC-positive FA, were based on a small sample size. Studies with small sample sizes are poorly representative, and their results may be relatively more unstable and more biased, leading to false positives. Finally, the lack of information did not allow for some subgroup analyses, such as gender and age analyses of the prevalence of SPT- and FA-positive FA. This was detrimental to understanding the factors that influence prevalence.

## 5. Conclusions

This meta-analysis confirmed that the prevalence of FA is common, varying by the assessment method. The pooled prevalence of self-reported FA and SPT-positive FA was higher than that of self-reported physician-diagnosed FA and OFC-positive FA, with an increasing trend over time. In addition, there was insufficient evidence to accurately predict the factors affecting the prevalence of FA, but age, region, and sample size appear to be important. These results will provide crucial informative value and baseline information for future investigations. Since the current studies on the prevalence of FA in China are mainly limited to specific ages and regions, there is an urgent need for a set of high-quality national epidemiological studies with uniform FA diagnostic criteria suitable for China to understand the true hazard profile of FA and produce a list of priority allergens food allergens in China. Moreover, it is expected to establish DBPCFC diagnostic methods to enrich FA epidemiological data in China and to obtain FA thresholds for Chinese people, providing a reliable scientific basis for FA management, prevention, treatment, and risk assessment in China.

## Figures and Tables

**Figure 1 nutrients-14-05181-f001:**
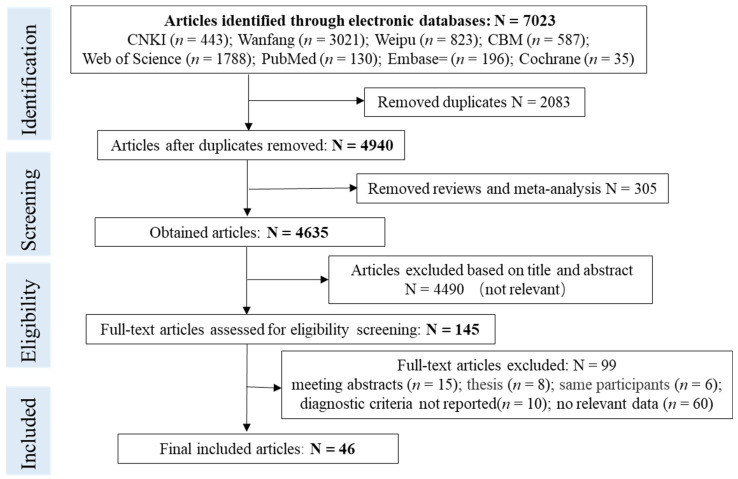
PRISMA flowchart for screening articles.

**Figure 2 nutrients-14-05181-f002:**
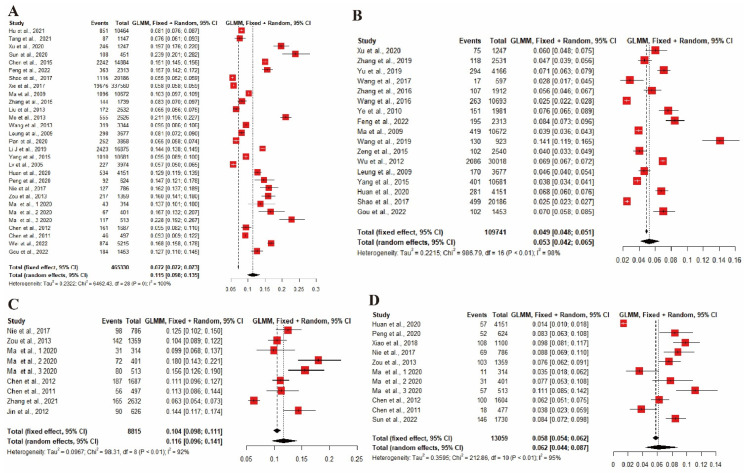
Forest plot of the pooled lifetime prevalence of FA in China. (**A**) Self-reported FA. (**B**) Self-reported physician-diagnosed FA. (**C**) SPT-positive FA. (**D**) OFC-positive FA. FA, food allergy; OFC, open food challenge; SPT, skin prick test [13,14,15,16,17,18,19,20,21,22,23,24,26,27,28,29,30,31,32,33,34,35,36,37,38,39,40,41,42,43,44,45,46,47,48,49,50,51,52,53].

**Figure 3 nutrients-14-05181-f003:**
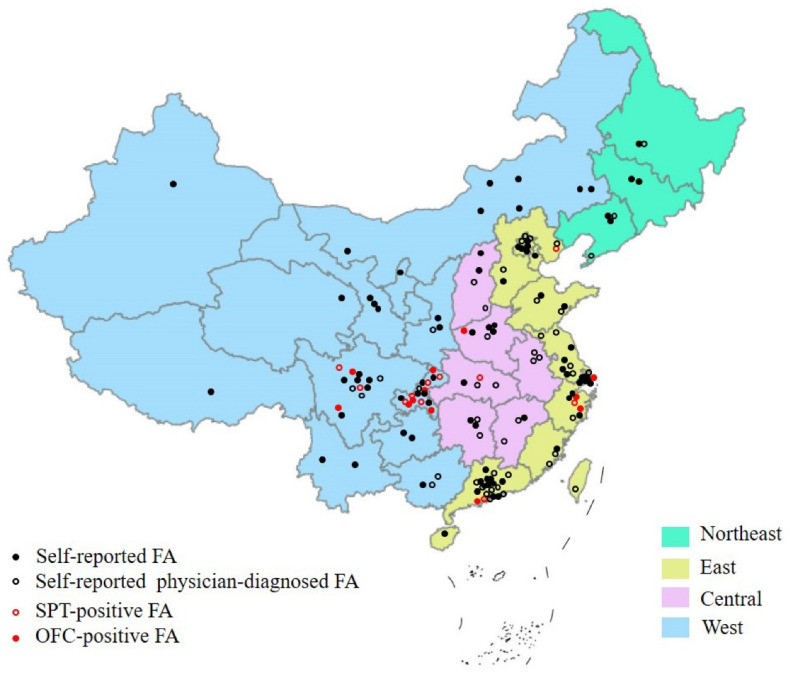
The geographic distribution of included studies on prevalence of FA in China. FA, food allergy; OFC, open food challenge; SPT, skin prick test.

**Table 1 nutrients-14-05181-t001:** Characteristics of included studies.

Study	Survey Region	Survey Period	Population Source	Age Range (years)	Sample Size	Diagnostic Method
Hu et al., 2021 [13]	Shanghai	2019.4–2019.6	School	6–11	10,464	Questionnaire
Tang et al., 2021 [14]	Shenzhen	2016.12–2017.3	School	10–16	1147	Questionnaire
Xu et al., 2020 [15]	Shanghai	2019.12–2020.1	School	0–5	1247	Questionnaire
Sun et al., 2020 [16]	Datong, Shanxi	2016.9–2016.12	School	around 16	451	Questionnaire
Zhang et al., 2019 [17]	Yangzhou, Jiangsu	-	School	3–6	2531	Questionnaire
Yu et al., 2019 [18]	Foshan, Guangdong	-	School	6–8, 12–14	4166	Questionnaire
Wang et al., 2017 [19]	Wuhu, Anhui	2014.11	Physical examination hospitals	0–2	597	Questionnaire
Zhang et al., 2016 [20]	Wuhu, Anhui	-	School	3–6	1912	Questionnaire
Wang et al., 2016 [21]	33 cities in China	2014.11	Physical examination hospitals	0–2	10,693	Questionnaire
Chen et al., 2015 [22]	Shanghai	2011.4–2012.4	School	3–7	14,884	Questionnaire
Ye et al., 2010 [23]	Wenzhou, Zhejiang	2009	School	16–25	1981	Questionnaire
Feng et al., 2022 [24]	Nanchang, Jiangxi	2019.12	School	18–24	2313	Questionnaire
Li et al., 2019 [25]	Beijing	2010	Community, School	0–14	13,073	Questionnaire
Shao et al., 2017 [26]	Beijing	2010	Community, School	0–15	20,186	Questionnaire
Xie et al., 2017 [27]	31 cities in China	2010	Community, School	0–14	337,560	Questionnaire
Ma et al., 2009 [28]	Beijing	2007.10–2008.1	School	6–11	10,672	Questionnaire
Wang et al., 2019 [29]	Chengdu, Sichuan	2014.3–2015.3	Cohort study	0–1	923	Questionnaire
Wang et al., 2018 [11]	Nei Mongol	2015.5–2015.8	Community	-	4441	Questionnaire
Zeng et al., 2015 [30]	Guangdong	2013.6–2013.12	School	1–7	2540	Questionnaire
Zhang et al., 2015 [31]	7 cities, 2 rural areas in China	2011.11–2012.4	School	3–12	1739	Questionnaire
Liu et al., 2013 [32]	8 cities in China	2011.10–2012.3	Physical examination hospital	0–3	2632	Questionnaire
Mo et al., 2013 [33]	Shanghai	-	School	15–20	2626	Questionnaire
Wang et al., 2013 [34]	YunnanGuizhouSichuan	2011.3–2011.7	Community	0.5–2	3344	Questionnaire
Wu et al., 2012 [35]	Taiwan	2004.4.1–2004.10.31	Physical examination hospital, Community, School	-	30,018	Questionnaire
Ho et al., 2012 [12]	Hong Kong	2005.9–2006.8	Community	0–14	7393	Questionnaire
Leung et al., 2009 [36]	Hong Kong	2006.11–2007.5	School	2–7	3677	Questionnaire
Zhang et al., 2021 [37]	Qinhuangdao, Hebei	2016.1–2017.1	Physical examination hospital	0–3	2632	Questionnaire, SPT
Pan et al., 2020 [38]	Wuxi, Jiangsu	2016.1–2017.12	School	3–14	3858	Questionnaire, sIgE
Li J et al., 2019 [39]	HongKong Guangzhou Shaoguan	2009.9–2016.6	School	7–10	16,875	Questionnaire, SPT, sIgE
Yang et al., 2015 [40]	Guangzhou Shaoguang	2010	School	7–12	10,681	Questionnaire, SPT, sIgE
Jin et al., 2012 [41]	Wuhan, Hubei	2011.6–2011.12	Physical examination hospital	0.5–0.75	626	Questionnaire, SPT
Lv et al., 2005 [42]	China Medical University	-	School	15–24	3974	Questionnaire, SPT
Huan et al., 2020 [43]	Wenzhou, Zhejiang	2018	School	3–6	4151	Questionnaire, SPT, sIgE, OFC
Peng et al., 2020 [44]	Sanmenxia, Henan	2017.1–2018.1	Physical examination hospital	0–1	624	Questionnaire, SPT, OFC
Xiao et al., 2018 [45]	Shanghai	2016.1–2017.6	Physical examination hospital	0–3	1100	Questionnaire, sIgE, OFC
Nie et al., 2017 [46]	Chengdu, Sichuan	2014.9–2015.3	Physical examination hospital	0–2	786	Questionnaire, SPT, OFC
Zou et al., 2013 [47]	Panzhihua, Sichuan	2010.1–2012.12	Community	0–3	1359	Questionnaire, SPT, OFC
Ma et al., 2020 [48]	Chongqing	1999/2009/2019	Physical examination hospital	0–2	314/401/513	Questionnaire, SPT, OFC
Chen et al., 2012 [49]	Chongqing Zhuhai Hangzhou	2009.1–2009.22010.1–2010.5	Physical examination hospital	0–2	1687	Questionnaire, SPT, OFC
Chen et al., 2011 [50]	Chongqing	2009.1.1–2009.2.28	Physical examination hospital	0–1	479	Questionnaire, SPT, OFC
Feng et al., 2017 [8]	Beijing	2012.5–2013.5	Physical examination hospital	39 ± 7	708	sIgE
Hu et al., 2015 [9]	National	2002	Serum bank	3–12	5190	sIgE
Hua et al., 2008 [10]	Xinjiang	2007.8–2007.12	Physical examination hospital	36–57	3067	sIgE
Wei et al., 2022 [51]	Shanghai	2019	School	3–6	5215	Questionnaire
Gou et al., 2022 [52]	Guizhou	2017–2018	School	18–20	1453	Questionnaire
Sun et al., 2022 [53]	Yiyang, Hunan	2018.1–2019.12	Physical examination hospital	0–3	1730	Questionnaire, OFC

FA, food allergy; OFC, open food challenge; SPT, skin prick test; sIgE, specific IgE—not available.

**Table 2 nutrients-14-05181-t002:** Subgroup analyses of the prevalence of FA in China.

Categories	Subgroups	No. of Studies	Sample Size (*n*)	Prevalence (% [95% CI])	Heterogeneity		*p* Across Subgroup
					I^2^	*p*	
Prevalence of self-reported FA	Gender						*p* = 0.81
	Male	11	199,963	10.7 (8.2–13.8)	100%	<0.01	
	Female	11	187,689	10.2 (7.8–13.3)	99%	<0.01	
	Age (years)						*p* < 0.01
	0–3	13	63,055	10.5 (7.7–14.2)	99%	<0.01	
	3–6	7	98,234	9.2 (6.4–13.1)	100%	<0.01	
	6–12	10	278,335	9.0 (7.3–11.1)	100%	<0.01	
	12–18	5	7843	12.0 (7.4–19.0)	99%	<0.01	
	>18	2	3766	14.2 (12.2–16.5)	-	-	
	Year of data collection						*p* = 0.02
	1999–2010	9	378,968	8.7 (6.8–11.0)	99%	<0.01	
	2011–2021	18	82,377	12.5 (10.4–15.0)	98%	<0.01	
	Regions						*p* < 0.01
	Northeast	2	31,984	7.4 (7.1–7.6)	66%	0.09	
	East	16	230,191	10.2 (8.3–12.5)	100%	<0.01	
	Central	4	55,203	9.9 (4.8–19.2)	99%	<0.01	
	West	11	139,926	12.1 (9.5–15.4)	99%	<0.01	
	Sample size						*p* = 0.02
	<1000	7	3586	16.3 (13.0–20.2)	89%	<0.01	
	1000–2000	6	8632	11.7 (8.8–15.4)	96%	<0.01	
	2001–5000	8	26575	9.9 (7.2–13.4)	99%	<0.01	
	>5000	8	426,537	10.0 (7.6–13.1)	100%	0	
Prevalence of self-reported physician-diagnosed FA	Gender						*p* = 0.43
	Male	8	24,561	4.6 (3.6–5.9)	95%	<0.01	
	Female	8	22,573	4.0 (2.9–5.4)	96%	<0.01	
	Age (years)						*p* < 0.01
	0–3	6	16,290	4.0 (2.1–7.4)	99%	<0.01	
	3–6	5	16,029	4.5 (3.3–6.1)	95%	<0.01	
	6–12	5	35,699	4.1 (3.0–5.6)	95%	<0.01	
	12–18	1	2971	7.0 (6.1–7.9)	-	-	
	>18	3	17,802	7.2 (6.2–8.2)	85%	<0.01	
	Year of data collection						*p* = 0.60
	1999–2010	5	66,534	4.7 (3.3–6.8)	99%	<0.01	
	2011–2021	9	34,598	5.4 (3.8–7.6)	98%	<0.01	
	Regions						*p* = 0.02
	Northeast	-	-	-	-	-	
	East	10	89,869	4.8 (3.9–5.8)	99%	<0.01	
	Central	2	2509	4.2 (2.6–6.7)	86%	<0.01	
	West	2	2376	10.0 (6.1–16.0)	97%	<0.01	
	Sample size						*p* < 0.01
	<1000	2	1520	6.6 (2.1–19.0)	98%	<0.01	
	1000–2000	4	6593	6.6 (5.8–7.4)	60%	0.06	
	2001–5000	6	19,378	5.7 (4.6–7.1)	93%	<0.01	
	>5000	5	82,250	3.6 (2.6–5.1)	99%	<0.01	
Prevalence of SPT-positive FA	Gender	-	-	-	-	-	-
	Age (years)	-	-	-	-	-	-
	Year of data collection						*p* = 0.75
	1999–2010	4	2889	12.3 (9.8–15.3)	81%	*p* < 0.01	
	2011–2021	4	4557	11.5 (8.0–16.2)	96%	*p* < 0.01	
	Regions						*p* < 0.01
	Northeast	-	-	-	-	-	
	East	2	3738	7.4 (5.8–9.6)	89%	*p* < 0.01	
	Central	1	626	14.4 (11.7–17.4)	-	-	
	West	7	4451	13.0 (11.2–15.1)	77%	*p* < 0.01	
	Sample size						*p* < 0.01
	<1000	6	3137	13.5 (11.6–15.6)	67%	*p* < 0.01	
	1000–2000	2	3046	10.8 (9.7–12.0)	0	*p* = 0.57	
	2001–5000	1	2632	6.3 (5.4–7.3)	-	-	
	>5000	-	-	-	-	-	
Prevalence of OFC-positive FA	Gender	-	-	-	-	-	-
	Age (years)						*p* < 0.01
	0–3	10	8908	7.4 (6.1–8.9)	77%	<0.01	
	3–6	1	4151	1.4 (1.0–1.8)	-	-	
	>6	-	-	-	-	-	
	Year of data collection						*p* = 0.45
	1999–2010	4	2796	5.3 (3.9–7.2)	69%	0.02	
	2011–2021	6	8904	6.8 (3.8–11.7)	97%	<0.01	
	Regions						*p* = 0.18
	Northeast	-	-	-	-	-	
	East	3	6305	4.3 (1.7–10.7)	99%	<0.01	
	Central	2	2345	8.4 (7.4–9.6)	0	0.93	
	West	7	4400	6.9 (5.3–9.0)	78%	<0.01	
	Sample size						*p* < 0.01
	<1000	6	3115	6.9 (4.9–9.5)	81%	<0.01	
	1000–2000	4	5793	7.9 (6.7–9.3)	83%	<0.01	
	2001–5000	1	4151	1.4 (1.0–1.8)	96%	<0.01	
	>5000	-	-	-	-	-	

FA, food allergy; OFC, open food challenge; SPT, skin prick test; sIgE, specific IgE—not available.

## Data Availability

Data is contained within the article or Appendix A.

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
