# Peer review of "Multi-Perspective Observation on the Prevalence of Food Allergy in the General Chinese Population: A Meta-Analysis"

_nutrients, 2022, doi:10.3390/nu14235181_

Round 1

Reviewer 1 Report

The manuscript “Multi-perspective observation on the prevalence of food allergy in the general Chinese population: a meta-analysis” by Wang J. and collaborators aims to estimate the prevalence of food allergy in the general Chinese population. The analysis was performed according to PRISMA’s criteria.    

This study is well performed.  The methods are appropiate for the objective of the study and the conclusions are supported by the results. 

Author Response

Comments and Suggestions for Authors

The manuscript “Multi-perspective observation on the prevalence of food allergy in the general Chinese population: a meta-analysis” by Wang J. and collaborators aims to estimate the prevalence of food allergy in the general Chinese population. The analysis was performed according to PRISMA’s criteria.    

This study is well performed.  The methods are appropiate for the objective of the study and the conclusions are supported by the results. 

Response: Thanks for your positive comments.

Reviewer 2 Report

The work of Wang et al. Chaykin et al. is a meta-analysis on the prevalence of food allergy in China, but only considering data from self-reported, self-reported physician-diagnosed, skin prick test-positive, and oral food challenge-positive allergies. The topic, despite the considerable number of reviews, continues to be interesting, and deserves attention and dedication by the scientific community. The work is in general well written. I don't have any major criticisms to make. Anyway, I emphasize that the work must be reviewed by someone with greater expertise in statistics because, although everything seems ok to me, the analyzes carried out must be validated. Important modifications must be conducted prior to possible publication.

Abstract – Please include the meaning of all the abbreviations (SPT, OFC…)

L25 – is common

L67 – have conducted

L86 – check the size of the text

L109 – were resolved

L112 - using t?

L112 – remove with

Figure 1 – in identification, several “;” are missing

L136 – 168 – please reformulate and explain better; hard to read

Figure 2 - the size of the individual figures (a-d) should be increased

L192 – Please explain “The prevalence of FA was significantly correlated with the sample size of the population surveyed” and “The prevalence of FA was significantly lower in large sample studies”.

L210 – italics?

L214 – 217 – ref. missing

L298 – was detrimental

L301 – is common; varying by the

L302 – “The pooled prevalence of self-reported FA and SPT-positive FA was higher than that of self-reported physician-diagnosed FA and OFC-positive FA, with an increasing trend over time.” I would like to see this subject better discussed. Also, it would be interesting to compare these results with those from double-blind placebo-controlled food challenges, even from other countries. It is not clear as is.

tive FA, with an increasing trend over time.

L311 -  allergens food allergens?

L311 – 315 – please reformulate; hard to read

Author Response

Comments and Suggestions for Authors

The work of Wang et al. Chaykin et al. is a meta-analysis on the prevalence of food allergy in China, but only considering data from self-reported, self-reported physician-diagnosed, skin prick test-positive, and oral food challenge-positive allergies. The topic, despite the considerable number of reviews, continues to be interesting, and deserves attention and dedication by the scientific community. The work is in general well written. I don't have any major criticisms to make. Anyway, I emphasize that the work must be reviewed by someone with greater expertise in statistics because, although everything seems ok to me, the analyzes carried out must be validated. Important modifications must be conducted prior to possible publication.

Response: Thanks for your comments. Actually, the manuscript has been reviewed by experts in epidemiology and statistics before submission. The manuscript has been revised in accordance with your suggestions.

Abstract – Please include the meaning of all the abbreviations (SPT, OFC…)

Response: Thanks for your suggestion. We have included the meaning of all the abbreviations (SPT, OFC…) in the abstract (L22).

L25 – is common

Response: We have revised it in L26.

L67 – have conducted

Response: We have revised it in L67.

L86 – check the size of the text

Response: Thanks for your reminding. We have standardized the font type of the text. (L86)

L109 – were resolved

Response: We have revised it in L109.

L112 - using t?

Response: Thanks for pointing this out. We have removed this redundant letter “t” in L112.

L112 – remove with

Response: We have removed “with” in L112.

Figure 1 – in identification, several “;” are missing

Response: Thanks for your kind reminding. We have double-checked Figure 1 and revised the errors.

L136 – 168 – please reformulate and explain better; hard to read

Response: As suggested, we have revised this section to make it easier to understand. (L136-138). The details are as follows.

L136-138: The two articles that only described the period/point prevalence of self-reported FA without providing lifetime prevalence data were also not applicable to the meta-analysis.

Figure 2 - the size of the individual figures (a-d) should be increased

Response: As suggested, we have resized each figure (A-D) in Figure 2.

L192 – Please explain “The prevalence of FA was significantly correlated with the sample size of the population surveyed” and “The prevalence of FA was significantly lower in large sample studies”.

Response: Thank you for your suggestion. We have rearranged this part (L192-197) and added the explanation in the discussion section (L300-301). The details are as follows.

L192-197: Regarding the prevalence of FA according to sample size, the prevalence of FA was significantly higher in studies with smaller sample size (sample size <1000; 16.3% (95% CI: 13.0-20.2%) for self-reported FA and 6.6% (95% CI: 2.1-19.0%) for self-reported physician-diagnosed FA), lower in large sample studies (sample size >5000; 10.0% (95% CI: 7.6-13.1%) for self-reported FA and 3.6% (95% CI: 2.6-5.1%) for self-reported physician-diagnosed FA).

L299-300: Studies with small sample sizes are poorly representative, and their results may be relatively more unstable and more biased, leading to false positives.

L210 – italics?

Response: Thanks for your reminding. Actually, “Sensitivity analysis and publication bias” is a subtitle, and we have put it in italics.

L214 – 217 – ref. missing

Response: Thanks for your reminding. We have added the relevant references.

L298 – was detrimental

Response: We have revised it in L302.

L301 – is common; varying by the

Response: We have revised it in L305.

L302 – “The pooled prevalence of self-reported FA and SPT-positive FA was higher than that of self-reported physician-diagnosed FA and OFC-positive FA, with an increasing trend over time.” I would like to see this subject better discussed. Also, it would be interesting to compare these results with those from double-blind placebo-controlled food challenges, even from other countries. It is not clear as is.

tive FA, with an increasing trend over time.

Response: Thanks for your suggestion. We have added the discussion section to the manuscript (L232-236, L253-255) for the relevant content you mentioned. The details are as follows.

L232-236: Self-report is known to overestimate the true prevalence of FA because these may also include food intolerances or toxicities. Moreover, a positive SPT only represents sensitization and not allergy, which may also exaggerate the prevalence of FA to some extent. Studies in Africa have also found high rates of food sensitization and self-reported food allergy.

L253-255: Although DBPCFC is the gold standard for food allergy diagnosis, OFC is traditionally more commonly used than DBPCFC because of the challenges and limitations of DBCFC in practical applications.

L311 -  allergens food allergens?

Response: Thanks for reminding us to realize this mistake. We have removed this redundant word “allergens” in L314.

L311 – 315 – please reformulate; hard to read

Response: As suggested, we have rewritten this section (L315-318) to make it more logical and easier to understand. The details are as follows.

L315-318: Moreover, it is expected to establish DBPCFC diagnostic methods to enrich FA epidemiological data in China and to obtain FA thresholds for Chinese people, providing a reliable scientific basis for FA management, prevention, treatment and risk assessment in China.
